# Plant Plasma Membrane Proton Pump: One Protein with Multiple Functions

**DOI:** 10.3390/cells11244052

**Published:** 2022-12-14

**Authors:** Adrianna Michalak, Anna Wdowikowska, Małgorzata Janicka

**Affiliations:** Department of Plant Molecular Physiology, Faculty of Biological Science, University of Wroclaw, Kanonia 6/8, 50-328 Wrocław, Poland

**Keywords:** plasma membrane H^+^-ATPase, pleiotropy, multitasking, plant physiology, pathogenesis, membrane protein, stress conditions, proton pump

## Abstract

In plants, the plasma membrane proton pump (PM H^+^-ATPase) regulates numerous transport-dependent processes such as growth, development, basic physiology, and adaptation to environmental conditions. This review explores the multifunctionality of this enzyme in plant cells. The abundance of several PM H^+^-ATPase isogenes and their pivotal role in energizing transport in plants have been connected to the phenomena of pleiotropy. The multifunctionality of PM H^+^-ATPase is a focal point of numerous studies unraveling the molecular mechanisms of plant adaptation to adverse environmental conditions. Furthermore, PM H^+^-ATPase is a key element in plant defense mechanisms against pathogen attack; however, it also functions as a target for pathogens that enable plant tissue invasion. Here, we provide an extensive review of the PM H^+^-ATPase as a multitasking protein in plants. We focus on the results of recent studies concerning PM H^+^-ATPase and its role in plant growth, physiology, and pathogenesis.

## 1. Introduction

The plasma membrane proton pump (PM H^+^-ATPase) is part of the P-type subfamily of ATPases, which share the common characteristic of forming a phosphorylated intermediate state during their catalytic cycle [1]. PM H^+^-ATPase is an integral plasma membrane protein found in plant and fungal cells, where it is engaged in exporting cytoplasmic protons to the apoplast using energy from ATP hydrolysis. The active export of protons by the plasma membrane proton pump creates the electrochemical proton gradient (also known as the protonmotive force, PMF) that energizes the plasma membrane and drives the secondary transport of numerous chemical compounds [2]. By reason of extruding protons to the apoplast and thus generating protonmotive force, PM H^+^-ATPase participates in many crucial physiological processes in plants, including maintaining ionic and intracellular pH homeostasis, nutrient uptake, growth, and stomata opening [3,4,5,6].

Structural analysis has shown that the PM H^+^-ATPase found in *Arabidopsis thaliana* is a single transmembrane polypeptide of approximately 100 kDa, with both N- and C-terminus located on the cytoplasmic side of the plasma membrane [7,8]. The most common model of PM H^+^-ATPase (Figure 1) suggests the presence of 10 transmembrane helices in its structure, with four domains characteristic of P-type ATPases—A, P, N, and R, which are exposed to the cytoplasmic side [1,9]. The actuator (A) domain includes the N-terminus of the protein along with a small loop located between helices two and three. Between helices four and five, a structure called a big loop consisting of jointed domains P and N have also been identified. Phosphorylation of the PM proton pump occurs in domain P, which contains the conservative sequence DKTGTLT with an aspartyl residue to be phosphorylated during the catalytic cycle. The nucleotide-binding (N) domain is the site of ATP-binding. The regulatory (R) domain found at the C-terminus of the PM proton pump has two regions of autoinhibition and a 14-3-3 protein binding site [10]. Studies have confirmed that the PM proton pump can function as a monomer; however, models of functional di- and hexamers have also been presented [11,12]. Considering its many key functions in plant physiology, the expression and activity of the PM proton pump are strictly regulated at transcriptional and posttranslational levels [13]. Studies have shown that, in many plant species, the PM proton pump is encoded by a multigene family. Twelve genes have been identified in the model plant *Arabidopsis thaliana* (*AHA1-12*) [14]. The presence of multiple genes encoding PM H^+^-ATPase has also been confirmed in other important plant species such as tobacco (*PMA1-9*), rice (*OSA1-10*), or cucumber (*CsHA1-10*) [9,15,16]. Among the posttranslational modifications, phosphorylation and dephosphorylation play a significant role in the regulation of PM H^+^-ATPase activity. Six phosphorylation sites in the R domain of the AHA2 isoform in *Arabidopsis* have been identified. One of the phosphorylation events occurs at the penultimate threonine residue (Thr-947 in AHA2). Phosphorylation of this residue forms a binding site for 14-3-3 proteins, stabilizes the upregulated state of the PM proton pump, and mitigates its autoinhibition [17,18]. Notably, it has been established that residues Thr-924 and Thr-946 assist in binding 14-3-3 protein to AHA2 [17,18,19]. The Thr-881 residue of AHA2 has also been identified as a site where phosphorylation leads to PM H^+^-ATPase activation [20]. Conversely, phosphorylation at Ser-899 or Ser-931 leads to the inhibition of AHA2 activity [19,21].

For years PM H^+^-ATPase has attracted the interest of plant physiology researchers due to its fundamental function in energizing plasma membranes. Continuous studies have shown that this one protein—plasma membrane proton pump—is indeed engaged in various physiological processes in plants, such as maintaining intracellular pH homeostasis, nutrient uptake, cell elongation, and stomatal aperture [3,4,5,6]. Given its multifunctionality and sensitivity to various environmental factors, PM H^+^-ATPase plays a crucial role in plant adaptation to biotic and abiotic stress conditions such as pathogens attacks, non-optimal temperature, salt stress, drought, and heavy metals [22,23,24,25,26]. PM H^+^-ATPase involvement in the regulation of processes occurring in plants is summarized in Table 1. Recently, PM proton pumps have become a matter of interest in agricultural studies concentrating on improving crops and engineering more resilient plant varieties. In this review, we focus on the pleiotropic effect and multiple functions of plant PM H^+^-ATPase in physiological processes and its role in adapting to biotic as well as abiotic stresses such as high salinity and drought. Furthermore, we discuss the possible agricultural applications of PM proton pump studies.

## 2. The Pleiotropic Effect of Plasma Membrane Proton Pump

The PM H^+^-ATPase molecular function of extruding protons to the apoplast, thus maintaining ΔpH and plasma membrane potential, establishes its multiple biological functions in plants. Given its vital role, it is understandable that the PM proton pump is encoded by a group of isogenes instead of one, as it would be difficult to manage broad expression patterns in different cell types under a variety of conditions. Previous studies have shown that PM H^+^-ATPase isoforms are not functionally identical and may exhibit tissue-specific expression patterns in various plant species [15,16,63]. However, it has been established that some PM proton pump isoforms display wide expression patterns at high levels throughout the plant life cycle and thus function as housekeeping genes [63]. BDue to the overlapping roles of the most broadly expressed isoforms in some cell types, the analysis of PM H^+^-ATPase single isogene loss-of-function mutants might not show significantly altered phenotypes compared to wild-type plants. The lack of phenotypic alteration in some PM proton pump loss-of-function mutants can be explained by the compensation of their defects by other isoforms. However, due to the unique expression patterns of specific isoforms, single gene loss-of-function mutations can cause a strong pleiotropic phenotype in specific cell types or under certain stress conditions [3,5,27]. Furthermore, studies have shown that overexpression of the most abundant isoforms of PM H^+^-ATPase leads to a pleiotropic phenotype that affects plant development and physiology [28,29].

*AHA1* and *AHA2* are the two most highly expressed and abundant isogenes encoding PM proton pumps in the model plant Arabidopsis. Double-knockdown *aha1* and *aha2* mutant plants are embryonic lethal, which confirms the vital role of these proteins in development and growth. Single *AHA1* or *AHA2* isogene loss-of-function mutants do not display growth or developmental alterations under optimal laboratory conditions. This observation supports the idea that a single functional isoform, *AHA1* or *AHA2*, is sufficient to maintain PMF. However, in conditions challenging protonmotive force, such as high potassium (100 mM KCl) and high external pH (pH 8.5), *aha2* mutant seedlings exhibit significantly decreased root growth. Such root length alterations were not observed in wild-type plants grown under the same conditions. In contrast, *aha2* Arabidopsis mutants cultivated in the presence of lithium or cesium are more immune to growth inhibition caused by these toxic cations compared to wild-type plants. This phenotype is strictly connected to *AHA2* dysfunction as the reduced uptake of these positively charged toxins is caused by lower membrane potential and reduced driving force [3]. Further studies on Arabidopsis seedlings proved that *aha2* knockout plants exhibit an altered hypocotyl phenotype. The quantification analysis of chlorophyll content in Arabidopsis seedlings showed that *aha2* mutants have a fourfold increase in chlorophyll concentration compared to wild-type plants. In addition, the hypocotyl length of wild-type plants and *aha2* mutants grown in the presence of 100 mM KCl was measured. This experiment showed that in the environment of high potassium concentration, the hypocotyl length of the *aha2* mutant was reduced, whereas such aberrations were not observed in wild-type plants. Alterations in hypocotyl length were not observed in either *aha2* mutants or wild type plants grown on the control medium. Thus, a decrease in *aha2* mutant hypocotyl length can be connected to membrane depolarization caused by high potassium concentrations [27]. It is worth mentioning that more recent studies, in contrast to what has been described previously, have provided evidence that the pleiotropic phenotype of *aha2* mutants can be observed in plants grown under normal growth conditions. The results of studies performed in 2018 by Hoffmann et al. [5] indicated that the growth of root hairs in *aha2* Arabidopsis mutants under normal growth conditions was increased compared to wild-type plants. In addition to root hair growth alteration, a decrease in the length of trichoblast cells contributing to shorter root length was observed, compared to wild-type plants, in the *aha2* mutant. Similarly, it was observed that under normal growth conditions, *aha2* loss-of-function mutants contribute to the reduction in cotyledon size [5].

Amongst knockout mutant analysis, several studies have investigated the influence of PM proton pump overexpression on plant phenotypes. *NpPMA4* is one of the most abundant and highly expressed PM H^+^-ATPase isoforms in *Nicotiana plumbaginifolia* [63]. To examine the effect of *NpPMA4* overexpression, tobacco plants were transformed with a vector carrying the *NpPMA4* gene encoding PM proton pump lacking the last 103 amino acid residues (ΔPMA4), corresponding to C-terminus autoinhibitory domain. This study has shown that the ΔPMA4 PM H^+^-ATPase isoform lacking the autoinhibitory domain is constitutively active. *ΔPMA4* mutant plants exhibit several developmental alterations compared to wild-type plants, such as bent leaves, twisted stems, paler petals, and male sterility [28]. Another study investigated the role of *AHA2* isoform overexpression in stomatal guard cells (GC) in Arabidopsis growth and physiology using a strong guard cells promoter (*GCI*). *GCI::AHA2* Arabidopsis plants have ~25% wider stomatal apertures, which open more quickly than the stomata of wild-type plants. As a result, transgenic Arabidopsis plants overexpressing *AHA2* in GC exhibited greater total flowering stems, and rosette leaves growth, as well as an increase in the number of rosette leaves. This phenotype alteration was induced by relatively high light intensity and was not observed in the wild-type plants. It is also worth mentioning that in *GCI::AHA2* Arabidopsis plants, significantly higher stomatal conductance and CO_2_ assimilation rate were observed compared to wild-type plants, which contributes to increased photosynthetic rate [29].

All of the above-mentioned studies show the importance of PM proton pump in plant development, growth, and various physiological processes. Although only one PM H^+^-ATPase isoform is modified, even when limited to specific cell types, its involvement in diverse and numerous processes in plants contributes to the production of a pleiotropic phenotype.

## 3. PM Proton Pump Engagement in Multiple Crucial Processes in Plants

### 3.1. Growth

The PM H^+^-ATPase is a key element of plant molecular mechanisms regulating cell elongation and growth. Auxins are one of the extensively studied phytohormones involved in a wide range of growth and developmental processes such as embryogenesis, organogenesis, and tropisms [64,65]. As growth regulators, auxins indirectly modulate PM H^+^-ATPase activity and manipulate H^+^ ion fluxes across the plasma membrane for regulating processes involved in cell elongation [30,31,66]. It has been established that auxin induces the extrusion of protons via PM H^+^-ATPase activation leading to apoplast acidification (pH approximately 4,5–6) [67]. The decrease in apoplast pH triggers the activation of proteins that mediate cell wall loosening by acting on the polysaccharide network [68]. Additionally, PM H^+^-ATPase activation induces plasma membrane hyperpolarization and triggers K^+^ ion influx leading to water uptake and cell turgor increase [32,69]. Both cell wall loosening and turgor increase are crucial factors for cell growth and are part of the Acid Growth Theory [30,32,67,70].

In shoots, the mechanism of growth promotion upon auxin perception is in agreement with the Acid Growth Theory. Takahashi et al. [30] proved that auxin application onto Arabidopsis hypocotyl segments increased AHA2 Thr-947 phosphorylation and 14-3-3 binding leading to its activation. It has been postulated that PM H^+^-ATPase activation is induced via the TIR1/AFB pathway of auxin nuclear signaling, which is based on the de-repression of auxin-induced genes at the transcription level [33]. The expression of several members of the SMALL AUXIN UP mRNA (SAUR) gene family is strongly and rapidly induced by auxin perception [71]. Additionally, it has been established that SAUR19-24 subfamily proteins are engaged in promoting cell expansion [72]. Further studies revealed that SAUR9, -10, -19, -40, and -72 negatively regulate the PP2C.D subfamily of 2C protein phosphatases to modulate PM H^+^-ATPase activity. PP2C.D proteins decrease PM H^+^-ATPase activity by indirect regulation of penultimate Thr residue dephosphorylation and, therefore, 14-3-3 proteins binding inhibition. Thus, SAUR proteins, by inactivation of PP2C-family phosphatases, promote an increase in PM proton pump activity, leading to hypocotyl cell growth via an acid growth mechanism [73].

Increased ATP hydrolytic activity was observed in auxin-treated roots. Furthermore, analysis of phospho-proteomics data revealed that Thr-947 in AHA2 was highly phosphorylated in roots treated with auxin [31]. Recent studies have described transmembrane kinases (TMKs) interacting with PM proton pumps as components of the auxin signaling pathway. Auxin-triggered TMKs activation leads to phosphorylation of the penultimate Thr residue and activation of PM H^+^-ATPases [31,74]. Notably, TMK1-triggered apoplast acidification is increased at lower auxin levels [31].

### 3.2. Stomata Opening

Stomata opening and closing regulations are vital in terms of gas exchange and the maintenance of photosynthetic processes. Stomata opening is triggered by PM H^+^-ATPase activation, leading to plasma membrane hyperpolarisation, massive ion influx, and, therefore, guard cell swelling [75]. Studies have shown that PM proton pump activation in guard cells is induced by the perception of blue light by protein kinases acting as blue light receptors expressed in guard cells, named phototropins (phot1 and phot2) [34,35]. Notably, it was observed that red light also induces penultimate Thr residue and activates PM H^+^-ATPase in whole leaves via photosynthesis-dependent mechanisms [76].

Blue light perception by phototropins induces its activation via autophosphorylation and initiates a signaling pathway that leads to stomatal opening [35,77]. Auto-activated phototropins directly phosphorylate BLUE LIGHT SIGNALING1 (BLUS1) protein kinase, which indirectly passes the signal to type 1 protein phosphatase (PP1) [36]. PP1 and its regulatory subunit PRSL1 mediate the light signal to PM H^+^-ATPase via yet unknown mechanism that triggers PM proton pump phosphorylation at the penultimate Thr residue [78,79,80,81]. Recently, it was revealed that membrane-localized type 2C protein phosphatase clade D (PP2C.D) members mediate the dephosphorylation of penultimate Thr residue in guard cells AHA2. The *pp2c.d6/9* double mutant exhibited an open stomata phenotype in the dark and delayed AHA2 dephosphorylation in guard cells after blue light illumination. Additionally, in plants overexpressing *PP2C.D9,* the stomatal opening is inhibited after light illumination [37]. However, the mechanism leading to PP2C.D proteins inhibition and PM H^+^-ATPase activation upon blue light perception is still unknown.

### 3.3. Mineral Uptake

The essential mineral elements required for plant growth and development are taken up from the soil by roots and then transported to upper organs via the vascular system. The uptake of various charged compounds is facilitated by specific transporters and channels. Many of those transporters are H^+^ symporters activated by extracellular acidification, therefore, require PMF generated by PM H^+^-ATPase to maintain their transport activity [1,82]. Here we present a few examples of PMF-coupled uptake of mineral compounds.

Nitrogen is taken up from the soil in inorganic forms, such as nitrate (NO_3_^−^) and ammonium (NH_4_^+^), as well as organic compounds (urea, amino acids, short peptides) [83]. NO_3_^−^ uptake in roots is mediated by transporters belonging to NPF (NRT1/PTR) and NRT2 families [38,83,84]. It has been revealed by electrophysiological studies that nitrate uptake by roots is an active process that occurs via the 2H^+^/NO_3_^−^ symport mechanism [38,85]. Additionally, it has been described that the ammonium transporter AMT1, localized in the plasma membrane of root cells, functions as an NH4^+^/H^+^ symporter [39]. Corresponding to its NH4^+^/H^+^ symporter activity, PM H^+^-ATPase-mediated apoplast acidification leads to stimulation of AMT1-mediated NH4^+^ transport [39,86].

Phosphorus can be absorbed by root cells in two inorganic forms—H_2_PO_4_^−^ and HPO_4_^2−^ via transporters belonging to the phosphate transporters family (PHT). PHT1 subfamily members of phosphate transporters are localized in the plasma membrane and mediate inorganic phosphate uptake from the soil via the H^+^ symport mechanism [82]. The stoichiometry of H_2_PO_4_^−^ and HPO_4_^2−^ symport across plasma membrane has been described as 2H^+^/1H_2_PO_4_^−^ or 4H^+^/1H_2_PO_4_^−^ and 3H^+^/1HPO_4_^2−^ [40,87].

Potassium ions are transported into root cells via two K^+^ electrochemical gradient-dependent transport systems characterized by their K^+^ affinity level [88]. A low-affinity K^+^ transport system mediated by the K^+^ inward-rectifying channel of the Shaker family AKT1 is dominant upon high levels of K^+^ external concentration and facilitates the passive influx of potassium ions [89]. Members of the KUP/HAK/KT transporter family, mainly HAK1 (in barley, rice, and pepper) and HAK5 (in Arabidopsis and tomato) are involved in high-affinity K^+^ uptake systems in roots in low external K^+^ concentration [90,91]. Electrophysiological studies have confirmed that in higher plants, high-affinity K^+^ uptake is mediated by the H^+^/K^+^ symport mechanism [92,93]. Additionally, it was observed that extracellular acidification mediated by PM H^+^-ATPase increased KUP/HAK/KT transport activity [41,93]. Thus, it has been concluded that K^+^ uptake in low K^+^ external concentration is an active process driven by plasma membrane potential created by PM H^+^-ATPase via H^+^/K^+^ symporters [94].

In higher plants, SULTR1 and SULTR2, which belong to the SULTR family, are the major transporters responsible for sulfur (SO_4_^2−^) absorption from the soil [95,96,97]. The SO_4_^2−^ uptake occurs against the electrochemical gradient of the plasma membrane and thus is an energy-driven process requiring PM H^+^-ATPase activity [42]. The SULTR1 and -2 transporters localized in the plasma membrane couple sulfate influx with co-transport of protons and therefore function as H^+^/SO_4_^2−^ co-transporters. Additionally, SULTR1 and -2 transporter activity are increased in lower external pH, which is in agreement with the consensus that the proton gradient is the driving force for SO_4_^2−^ uptake in plants [98].

### 3.4. Cytosolic pH Homeostasis

The cytoplasmic pH (pH_cyt_.) of plant cells is strictly regulated and stabilized in the narrow range of 7.1–7.5 pH [43]. pH_cyt_. stabilization can be achieved via PM H^+^-ATPase activity which converts chemical energy stored in ATP into an electrochemical H^+^ gradient across the plasma membrane to fuel further secondary transport of charged compounds. It has been proven that secondary cation/H^+^ antiporters belonging to NHX (Na^+^/H^+^ exchanger) and CHX (cation/H^+^ exchanger) families play a major role in establishing pH_cyt_. in normal as well as stress conditions [43,44,99]. Moreover, intercellular metabolic processes, either consuming or producing H^+^, are a vital part of a cytosolic pH stabilization mechanism, as they prevent alkalinization of the cytosol during H^+^ efflux [45,100]. Studies have shown that PM H^+^-ATPase activity is sensitive to changes in the pH of cytoplasm and achieves its maximum at approximately 6.5–6.8 pH [101,102]. Additionally, the generation of H^+^ ions during metabolic processes provides PM H^+^-ATPase with its substrate during long-term H^+^ fluxes [46].

### 3.5. Adaptation to Salt Stress

One of the major environmental abiotic stress factors that globally affects plant growth and physiology is highly saline soil [103]. High concentration of salt in the soil causes the accumulation of toxic levels of Na^+^ in plant cells, leading to osmotic stress and disruption of cellular ion homeostasis [104]. To overcome salt stress, plants have evolved a wide range of adaptation mechanisms by perceiving high Na^+^ concentrations and changes in osmotic pressure [105,106]. Among various molecular pathways initiated during salt stress, PM H^+^-ATPase is one of the key elements in osmoregulation and ion homeostasis maintenance [47]. Studies have shown that PM H^+^-ATPase is upregulated at the transcriptional and posttranslational levels in high salinity conditions [48,49,107]. It has been established that maintaining an optimal intercellular K^+^/Na^+^ ratio is essential for salt stress tolerance in plants [108,109,110]. To sustain a low Na^+^ level in the cytoplasm in a highly saline environment, Na^+^ extrusion to the apoplast is mediated by Na^+^/H^+^ antiporter identified as SOS1 (Salt Overly Sensitive 1) [111]. The SOS1 transporter is a part of the SOS regulatory pathway activated under salt stress conditions [50,112,113]. The SOS1 Na^+^/H^+^ antiporter requires an electrochemical proton gradient generated by PM H^+^-ATPase for the extrusion of Na^+^ against its electrochemical gradient [47]. Additionally, it has been observed that the plasma membrane is depolarized under saline conditions due to massive Na^+^ influx [114]. The most severe effect of PM depolarization for ion homeostasis in plant cells is drastic K^+^ efflux via outward-rectifying depolarization-activated (GORK) channels [51,114]. It was reported that plant varieties with intrinsically higher activity of PM H^+^-ATPase displayed lower NaCl-induced K^+^ efflux due to more negative resting plasma membrane potential [51].

### 3.6. Adaptation to Drought Stress

Plant response to drought conditions involves mechanisms that lead to stomata closure during the daytime to minimize water loss due to transpiration [75]. Studies have shown that the abscisic acid (ABA) phytohormone signaling pathway regulates PM H^+^-ATPase activity in guard cells under drought stress [52,53]. ABA-mediated stomata closure defense mechanism upon drought sensing relies on plasma membrane depolarization due to activation of anion efflux channels, which leads to K^+^ efflux from guard cells via K^+^ outward-rectifying channel. Additionally, ABA inhibits the activity of K^+^ inward-rectifying channels [115]. Taken altogether, sustained K^+^ efflux from guard cells leads to the loss of guard cell turgor and stomata closing [116]. To maintain plasma membrane depolarization and stomata closing under drought conditions, ABA negatively regulates PM H^+^-ATPase activity in guard cells by dephosphorylation of penultimate Thr residue [117,118]. It has been suggested that PM H^+^-ATPase inactivation in ABA-mediated pathway stomata closure is triggered by members of the SnRK2 kinase family [54,117,118].

### 3.7. Heavy Metals and Temperature Stresses—Poststress Responses for Homeostasis Maintenance

Some heavy metals, such as Zn and Cu, are essential elements for plant growth, while others, such as Cd, are not. Nevertheless, they are toxic to plants at high concentrations. Damage to the cell membrane system, especially the plasma membrane, is one of the primary events of heavy metal toxicity in plants. Disruption of membrane integrity is thought to be due to complex interactions between heavy metals and the functional groups of membranes. It is well known that metal ions are easily bound to both the sulfhydryl groups of proteins and the hydroxyl groups of phospholipids [119]. They can also replace calcium ions at essential sites on the cell membranes [120]. All these events result in an increase in non-specific membrane permeability and a parallel decrease in specific transport activities that disrupt ionic homeostasis and, subsequently, the activities of many enzymes crucial for basic cell metabolism.

The effect of metals on PM H^+^-ATPase activity depends on the time of exposure of plants to heavy metals, the type and concentration of heavy metals, and plant species. In cucumber seedling roots, a two-h treatment with Cd or Cu (10 and 100 μM) inhibited PM H^+^-ATPase activity [23]. Kennedy and Gonsalves [121], Fodor et al. [56], and Burzyński and Kolano [57] observed a similar inhibitory effect of short-term treatment with Cd or Cu on the activity of proton pumps in the roots of different plants. However, a longer treatment time (six days) of plants (cucumber) with heavy metals (Cd and Cu) led to increased activity of the enzyme [122]. Moreover, the expression of a few isogenes encoding PM H^+^-ATPase in Cd-treated cucumber seedlings was upregulated. In contrast, alteration of PM proton pump activity in cucumber roots stressed with Cu did not appear to affect gene expression levels [122]. Similarly, in rice treated for five or ten days with Cd, increased proton pump activity has been observed [123]. Hippler et al. [124] found that treating plants with copper for 72 h inhibited *AHA2* expression in *Arabidopsis thaliana* but had no effect on *AHA1* and *AHA5*. *AHA2* is the predominant proton pump in the roots and is upregulated after nitrate supply [125]. In contrast, treatment of the same plants with Cu for a longer period (15 days) did not inhibit *AHA2* gene expression [124].

When plants are exposed to heavy metal stress for a long time, they must replenish lost nutrients and remove toxic excess heavy metals from the cytosol to survive. Maintaining the active transport of ions and organic compounds across the PM requires the increased generation of a proton gradient by PM H^+^-ATPase. Besides using the proton gradient to replenish the loss of essential substances in repair processes, the more important to plants is to remove excess toxic ions from the cytoplasm to the outside of cells. In plants, transporters of the cation diffusion facilitator (CDF) family appear to mediate the cytoplasmic efflux of heavy-metal cations. CDF family proteins are membrane-divalent cation transporters that transfer metal ions out of the cytoplasm into the extracellular space or vacuoles, and they act as metal^2+^/H^+^ antiporters [126]. These proteins are known as metal-tolerance proteins (MTPs). In this family, at least one MTP8 could participate in the efflux of Cd and Cu from the cytosol, as it has been shown that *MTP8* is upregulated in response to Cd and Cu in the roots of *Arabidopsis halleri* [127]. The mechanism of Cd detoxification that relies on Cd^2+^/H^+^ antiport activity in plant plasma membranes has been previously reported [57].

Similar to what has been observed regarding heavy metal stress, low- and high-temperature conditions disrupt plasma membrane integrity. The biophysical lipid properties of the plasma membrane contribute to its sensitivity to temperature changes [128]. Both low and high temperatures modulate plasma membrane fluidity [129]. During exposure to low-temperature conditions, the proportion of unsaturated fatty acids in the plasma membrane increases, leading to rigidification. Conversely, high temperatures contribute to membrane fluidization [129]. Alterations in plasma membrane fluidity may affect the activity of proteins localized within their structure [55]. Additionally, electrolyte leakage caused by increased plasma membrane permeability was observed under both low- and high-temperature conditions [55,130,131]. Therefore, plants exposed to non-optimal temperature conditions survive, activate repair mechanisms that restore ionic homeostasis and replenish the loss of essential compounds [129,132,133]. PM H^+^-ATPase is the key enzyme in maintaining ion homeostasis in plants, considering its role in generating protonmotive force and thus energizing the secondary transport of numerous chemical compounds, as described in previous sections. Therefore, multiple studies have investigated the expression and activity patterns of PM H^+^-ATPase in plants exposed to temperature-stress conditions.

In cucumber seedlings, decreased hydrolytic and transporting activity of PM H^+^-ATPase was observed in plants transferred to low temperatures (10 °C) for two and three days, compared to seedlings grown under control conditions (25 °C) [24]. Notably, the expression of the most abundant PM H^+^-ATPase isogenes in the roots of cucumber grown for three days at 10 °C was lower than that in the control plants. However, a significant increase in the activity of cucumber PM H^+^-ATPase was detected in plants grown for five, six, and seven days at 10 °C, as well as in plants grown at 10 °C for three days and then transferred to 25 °C for the following three–four days (post-stress plants). Additionally, the analysis of PM H^+^-ATPase cucumber isoform transcripts showed that after six days at 10 °C, the expression of a few *CsHA* genes was significantly increased [24]. Parallel analysis was conducted on Arabidopsis seedlings grown under low-temperature conditions (4 °C) [55]. A strong decrease in the activity of PM H^+^-ATPase in Arabidopsis seedlings after six hours of exposure to 4 °C was observed compared with the control group. The activity of PM H^+^-ATPase in Arabidopsis seedlings recovered after 12 h of exposure to low-temperature conditions. Notably, 18 h of exposure to 4 °C caused a significant (approximately four-fold) induction of PM H^+^-ATPase activity in comparison to seedlings in the control group. Exposure of Arabidopsis seedlings to low temperatures for 12 h resulted in a substantial increase in *AHA1* and *AHA2* transcription. Remarkably, after 18 h of cold stress, the expression of *AHA1* and *AHA2* was up to 100-fold higher than that in the control plants [55].

Data regarding PM H^+^-ATPase activity and expression patterns under high-temperature conditions are currently limited. However, a study of cucumber seedlings exposed to heat shock (HS) showed stimulation of PM H^+^-ATPase transport and hydrolytic activity after two hours of incubation at 48 °C, compared to plants grown under control conditions (25/22 °C). Corresponding results were obtained for cucumber seedlings transferred to control conditions for 24 h after 2 h of exposure to HS (post-stress plants). Real-time PCR revealed an increase in the expression levels of *CsHA4* and *CsHA8* in post-stress plants compared to control plants [134].

These studies presumably indicate that PM H^+^-ATPase initially undergoes inactivation during temperature stress due to a plasma membrane integrity disorder. However, its activity is subsequently restored. The recovery of PM H^+^-ATPase activity during ongoing temperature stress could be a part of the plant adaptation mechanisms to unfavorable temperature conditions. The increased expression levels of the most abundant isoforms of PM H^+^-ATPase in plants treated with high or low temperatures indicate the participation of transcriptional pathways in the adaptation process. Additionally, studies have shown that PM H^+^-ATPase is highly activated during the post-stress stage. Taken together, these observations imply that PM H^+^-ATPase is a crucial element for ionic homeostasis restoration as well as a vital factor for replenishing the loss of vital compounds due to an increase in PM permeability during temperature stress. Nonetheless, further studies on PM H^+^-ATPase activity and regulation of expression under temperature stress are required.

## 4. PM Proton Pump Involvement in Plant Pathogenesis

To date, numerous studies improved our understanding of the molecular mechanisms underlying the unique, innate immunity of plants. Plant immunity is based on two pathogen recognition mechanisms. One of them relies on intracellular resistance proteins (R), which activate effector-triggered immunity (ETI) upon specific virulence factor recognition [135,136]. The second mechanism is based on the recognition of conserved microbial features called pathogen-associated molecular patterns (PAMPs) by extracellular pattern recognition receptors (PRRs). Binding PAMP to specific PRR activates pattern-triggered immunity (PTI) responses, including mitogen-activated protein kinase (MAPK) signaling, an increase of cytosolic Ca^2+^ levels, reactive oxygen species (ROS) and hormone production, as well as pathogenesis-related (PR) gene expression [135,136]. It was also proven that PTI responses induce increased callose deposition and stomatal closure, which form a physical barrier preventing the systemic spread of pathogens [135]. It is a well-known fact that stomata function in plants as pores enabling gas exchange; however it has been observed that they can be targeted by some types of pathogens as entry portals [58]. Plants evolved mechanisms that induce stomata closure upon PAMPs recognition to prevent pathogen internalization, yet several pathogens overcome stomatal defense mechanisms by secreting various chemical compounds triggering stomatal reopening [137]. Since PM H^+^-ATPase is directly involved in regulating stomatal close/open state, its role in PTI during pathogenesis has been investigated.

Widely studied, the interaction between the pathogen *Pseudomonas syringae* pv *tomato* (*Pto* or *Pst*) and its host’s tomato (*Solanum lycopersicum*) and the model plant *Arabidopsis thaliana* helped us better understand molecular mechanisms underlying plant pathogenesis and PM H^+^-ATPase involvement in those processes [138]. Studies have shown that bacterial flagellin presenting the flg22 epitope is recognized by Arabidopsis PRR called FLS2 (Flagellin Sensitive 2) [135]. The FLS2 perception of flg22 triggers a cascade of reactions. One of the earliest defense responses includes changes in PM proton pump phosphorylation status at three sites at the autoinhibitory C-terminal domain leading to PM H^+^-ATPase inactivation [25]. The PM proton pump inactivation is accompanied by Cl^−^ and K^+^ effluxes, as well as H^+^ influx across the PM, leading to PM depolarization and apoplast alkalization required for stomata closure [139]. Within one hour of PAMP recognition, stomatal closure is induced [58]. AHA1 and AHA2 inactivation event is crucial for stomata re-opening inhibition [137].

RIN4 (RPM1-interacting protein 4) is a widely studied plant innate immunity regulator. It has been observed that RIN4 interacts with the C-terminal autoinhibitory of AHA1 and AHA2 and positively regulates its activity [60]. The mechanism of PM proton pump activation mediated by RIN4 protein is yet to be discovered. The results of the 2017 study showed that the GCN4 (GENERAL CONTROL NONREPRESSIBLE4) protein, part of the AAA+-ATPase proteins family, is involved in plant immunity mechanisms as well. Kundal et al. observed that GCN4 interacted directly with RIN4 and 14-3-3 proteins. Furthermore, RIN4 and 14-3-3 proteasome-mediated pathway degradation was observed upon GCN4 overexpression and pathogen infection. The degradation of the RIN4/14-3-3 complex during a pathogen attack triggers PM H^+^-ATPase inactivation for stomata re-opening and pathogen internalization prevention. Kundal et al. [61] speculated that endogenous levels of GCN4 are not sufficient to trigger RIN4/14-3-3 degradation; however, the results of their study showed that GCN4 expression was induced during a pathogen attack. Thereby, GCN4-RIN4-14-3-3 protein complex stoichiometry is presumably a vital factor for RIN4/14-3-3 degradation and, thus, PM H^+^-ATPase inactivation [61].

However, just a few hours after infection, *Pto* secretes many effector proteins capable of PTI suppression [140]. *Pto* effectors such as AvrB and AvrRpm1 target RIN4 protein and manipulates its phosphorylation status leading to AHA1 and/or AHA2 re-activation. The activation of PM H^+^-ATPase leads to the hyperpolarization of the plasma membrane and induction of inward K^+^ efflux. Those events contribute to turgor increase in guard cells and stomatal reopening [60]. Furthermore, it has been proven that AHA1 re-activation upon AvrB secretion promotes the jasmonate (JA) signaling pathway, which is required for efficient stomatal infection upon pathogen attack [62].

*AHA1*, *AHA2*, and *AHA5* are PM proton pump isoforms majorly expressed in *Arabidopsis thaliana* guard cells [141]. The role of AHA1 and AHA2 in stomatal-defense mechanisms under *Pto* pathogen attack is well established [58,60]. Until recently, the potential role of AHA5 in plant innate immunity was not described. In 2022 Zhao et al. [59] provided evidence that AHA5 is indeed involved in PTI mechanisms employed against *Pto* attack. AHA5, presumably, is involved in H_2_O_2_ increased production during PTI. Additionally, AHA5, similarly to AHA1 and AHA2, forms a complex with RIN4 protein, however, further interactions between AHA5 and RIN4 during pathogenesis are yet to be established [59].

## 5. Future Perspectives

The PM proton pump generates the protonmotive force required for plasma membrane energization and driving secondary transport, thus functioning as a regulatory center of development, growth, and physiological processes in plants. The PM H^+^-ATPase multifunctionality in plants is summarized in Figure 2. Indeed numerous studies have shown that the abundance of PM H^+^-ATPase in plants is correlated with its pleiotropic effect. The phenomenon of pleiotropy has become the basis for the development of a new strategy for crop yield improvement by modulating the expression of a single gene encoding PM H^+^-ATPase. Zhang et al. 2021 examined the phenotype of OSA1-overexpression transgenic rice plants driven by the CaMV-35S promoter (OSA1-oxs) and compared them with wild-type plants and *osa1* knockout mutants. OSA1-ox plants showed increased PM H^+^-ATPase activity and enhanced stomatal conductance and photosynthetic activity compared to control plants, which enhanced carbon fixation and increased the biomass of OSA1-ox plants. Furthermore, an improved ability for NH_4_^+^, Pi, and K^+^ uptake in OSA1-oxs was observed, in contrast to the *osa1* knockout mutants with a lower uptake of nutrients. At the same time, the expression levels of certain transporter genes, such as high-affinity Pi, NH_4_^+^ transporters, and high-affinity K^+^ transporter HAK1, were upregulated in OSA1-ox plants. These results confirm that nutrient uptake that take place in roots is highly dependent on PM proton pump activity and that PM H^+^-ATPase plays a central role in rice yield enhancement. The authors took a step forward and investigated OSA1-ox rice yields for two growing seasons in different locations in China [142]. As expected, the growth and grain yield of OSA1-oxs were approximately 33% higher than those of the control plants and significantly lower in the case of *osa1* mutants at all examined locations. What is extremely interesting from the point of view of agricultural production is that OSA1-ox grain yield was still higher than that of control plants, even when treated with half the typical level of N fertilizer [142]. Experiments indicate that OSA1-ox rice can increase nitrogen more effectively than the control plants. This is an extremely important observation considering that the use of large amounts of fertilizers places a heavy burden on ecosystems and agriculture [143]. OSA1-ox rice and other plants overexpressing PM proton pumps have gained the name PUMP plants—the promotion and upregulation of plasma membrane proton-ATPase [142]. PUMP plants, especially those obtained using non-transgenic methods, such as Crispr/cas9, could have great potential for future commercial use, yielding higher yields and environmental benefits.

Another possible application of plant PM H^+^-ATPase in sustainable agriculture is its use as a marker of biostimulant effects [144]. Since the PM proton pump is essential for plant growth and development and is a pivotal enzyme for the transport mechanism in plant cells, it has been indicated that growth regulators can affect it [145]. Therefore, plant PM H^+^-ATPase has been presented as a potential biochemical marker of biostimulant activity, particularly for the stimulation of plant root growth. Protocols for testing biostimulants, such as humus or other derived substances with auxin-like activity, have already been proposed [144], and are based on determining the degree of root acidification. However, the vast majority of natural ingredients have an indirect effect on the proton pump, and additional researches, more specifically, are necessary to investigate the potential of the plant PM proton pump as a target for biostimulants or other biologicals [145].

## 6. Conclusions

Numerous studies throughout the decades discovered multiple functions of PM H^+^-ATPase in plants. The significance of PM proton pump for plant development, growth, and regulation of many physiological processes is well established. The complexity of PM H^+^-ATPase regulation on transcriptional and posttranslational levels provides proof of its immense potential in managing various processes in response to multiple stimuli in different types of cells. Phosphorylation is pivotal posttranslational modification regulating PM H^+^-ATPase activity. However, further studies on PM H^+^-ATPase molecular regulatory mechanisms are still required. Most importantly, the kinase responsible for the most crucial phosphorylation for PM proton pump activity at the penultimate Thr residue is yet to be identified. Discovering new sites of phosphorylation and further identifying kinases targeting phosphosites is essential for understanding molecular mechanisms underlying PM proton pump involvement in physiological processes in plants. The insight into responses to various unfavorable environmental conditions is crucial for revealing mechanisms leading to plant adaptation. Close attention should be given to the molecular pathways leading to PM H^+^-ATPase regulation under abiotic stresses such as drought and high salinity, which may help invent new, more resilient plant varieties.

## Figures and Tables

**Figure 1 cells-11-04052-f001:**
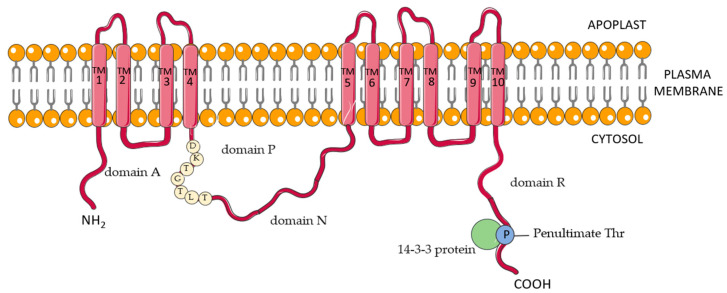
Schematic structural model of plasma membrane (PM) H^+^-ATPase in the plant cell. Characteristic domains and motifs for the PM proton pump are discussed in the text. The figure was made using Servier Medical Art. Database (SMAD online, available at https://smart.servier.com, accessed on 17 November 2022) and made available under the terms of the Creative Commons Attribution 3.0 license.

**Figure 2 cells-11-04052-f002:**
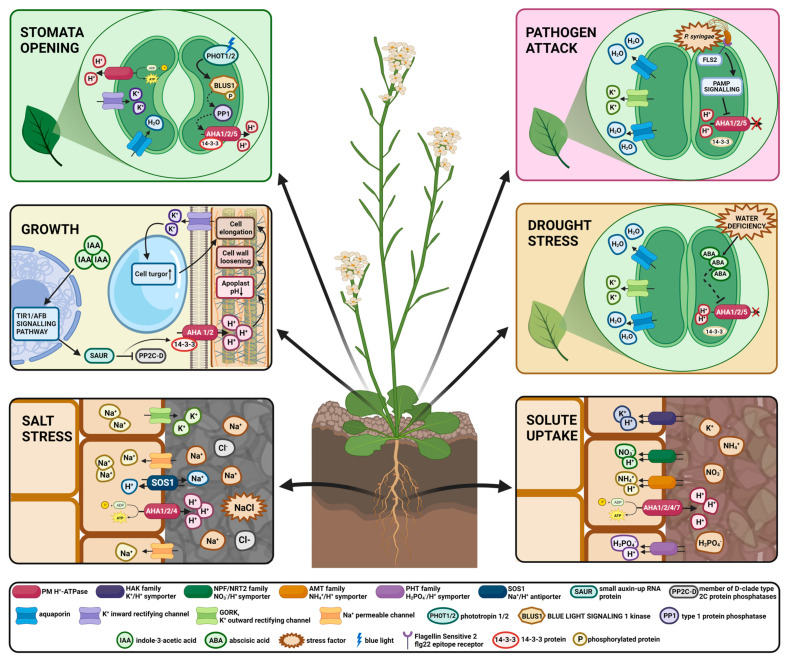
Summary of PM H^+^-ATPase multitasking in physiological processes in *Arabidopsis thaliana*. The figure was created with BioRender.com (Available at https://biorender.com, accessed on 12 December 2022).

**Table 1 cells-11-04052-t001:** The processes in plants are regulated by PM H^+^-ATPase activity.

Process	References
Development	Haruta et al. 2010 [3], Hoffmann et al. 2019 [5], Haruta et al. 2012 [27], Gévundant et al. 2007 [28], Wang et al. 2014 [29]
Growth of root and shoot	Takahashi et al. 2012 [30], Li et al. 2021 [31], Hager 2003 [32], Fendrych et al. 2016 [33]
Physiological Processes	
Stomata opening	Kinoshita et al. 1999 [34], Kinoshita et al. 2002 [35], Talemiya et al. 2016 [36], Akiyama et al. 2022 [37]
Mineral compounds uptake	Espen 2004 [38], Ortiz-Ramirez et al. 2011 [39], Młodzińska and Zboińska 2016 [40], Maathius et al. 1993 [41], Godwin et al. 2003 [42]
pH homeostasis maintenance	Sze and Charnoj 2018 [43], Bassil and Blumwald 2014 [44], Wegner et al. 2021 [45], Wegner and Shabala 2020 [46]
Adaptation to Abiotic Stresses	
Salt stress	Janicka-Russak and Kabała 2015 [47], Yang et al. 2019 [48], Janicka-Russak et al. 2013 [49], Qiu et al. 2002 [50], Chen et al. 2007 [51]
Drought	Goh et al. 1996 [52], Zhang et al. 2004 [53], Yin et al. 2013 [54]
Low temperature	Janicka-Russak et al. 2012 [24], Muzi et al. 2016 [55]
Heavy metals	Janicka-Russak et al. 2008 [23], Fodor et al. 1995 [56], Burzyński and Kolano 2003 [57]
Pathogenesis	
Plant defense mechanism	Falhof et al. 2016 [25], Melotto et al. 2006 [58], Zhao et al. 2022 [59]
Pathogen attack	Liu et al. 2009 [60], Kundal et al. 2017 [61], Zhou et al. 2015 [62]

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
