# Peer review of "Plant Plasma Membrane Proton Pump: One Protein with Multiple Functions"

_cells, 2022, doi:10.3390/cells11244052_

Round 1
Reviewer 1 Report
The manuscript has thoroughly looked through the functions of PM H+-ATPase, a very important field. It clear and comprehensive.
Some minor comments:
-Abstract can be more focused and crisp.
-Just a suggestion – maybe summarize the known PM H+-ATPase genes (involved in plant physiological processes, adaptation to abiotic stresses, and pathogenesis), roles, and organize it a table. The table would help and make this article a nice resource.
-Also, adding a section describing the roles of PM H+-ATPase in adaptation to low temperature and heavy metals, as that to salt stress and drought.
Author Response
Thank you very much for revision of our manuscript. We follow carefully the comments and, according to them, improved our paper.
Some minor comments:
- Abstract can be more focused and crisp.
RESPONSE: Dear Reviewer, we appreciate your efforts for reviewing the paper and providing valuable comments. We have rewritten abstract to be more focused and crisp.
- Just a suggestion – maybe summarize the known PM H+-ATPase genes (involved in plant physiological processes, adaptation to abiotic stresses, and pathogenesis), roles, and organize it a table. The table would help and make this article a nice resource.
RESPONSE: We have addressed suggestion given by Reviewer and we have prepared the figure entitled “Summary of PM H+-ATPase multitasking in physiological processes in Arabidopsis thaliana”. New figure includes AHA isoforms in Arabidopsis involved in mentioned processes. We hope that this figure makes indeed this article a nice resource.
- Also, adding a section describing the roles of PM H+-ATPase in adaptation to low temperature and heavy metals, as that to salt stress and drought
RESPONSE: We want to thank the Reviewer for this comment. We have added a section describing the roles of PM H+-ATPase in adaptation to extreme temperature as well as heavy metals.
Reviewer 2 Report
In this review, Michalak et al. summarize and describe the various functions of the PM H+-ATPase in plants. Overall the writing, phrasing and grammar of the manuscript are excellent and understandable. Very well done. Only some corrections and rephrasing have to be made in the manuscript itself (see minor points for the text).
However, one additional Figure would greatly help to increase the understanding of this article (see major points for the text).
If the points mentioned in detail below can be addressed by the authors in a revision, this already great review is ready for publication and will be a helpful contribution to the field. Although there are only small corrections, the addition of another Figure is a little bit more work, so I will demand a major revision to see the improvement of this article before publication. This article represents high quality scientific writing. Very looking forward to future articles about the topic.

Author Response
First of all we would like to thank you for detailed revision of our manuscript. We followed carefully the comments and improved our paper.
"In this review, Michalak et al. summarize and describe the various functions of the PM H+-ATPase in plants. Overall the writing, phrasing and grammar of the manuscript are excellent and understandable. Very well done. Only some corrections and rephrasing have to be made in the manuscript itself (see minor points for the text).
However, one additional Figure would greatly help to increase the understanding of this article (see major points for the text).
If the points mentioned in detail below can be addressed by the authors in a revision, this already great review is ready for publication and will be a helpful contribution to the field. Although there are only small corrections, the addition of another Figure is a little bit more work, so I will demand a major revision to see the improvement of this article before publication. This article represents high quality scientific writing. Very looking forward to future articles about the topic."
RESPONSE: Dear Reviewer, thank you for reviewing the manuscript and providing valuable suggestions. We performed revision as followed:
"Major points for the text:
As mentioned above, one additional Figure, which displays the various functions of the PM H+-ATPase at the level of the plasma membrane would be of great help for understanding this article. Moreover, only one Figure for a review article is a little bit too less. So basically, make a Figure, which depicts the different functions in growth, stomata opening, mineral uptake, cytosolic pH regulation, salt stress adaption and one example of plant immunity against a pathogen of your choice."
RESPONSE: Thank you for your valuable attention. Obviously we agree with the opinion of Reviewer and we have prepared additional figure entitled “Summary of PM H+-ATPase multitasking in physiological processes in Arabidopsis thaliana” which depicts the different functions of plasma membrane proton pump in plants.
"In lines 390-412: This passage is way too detailed for the conclusion. It is well written but belongs in some part of the main text."
RESPONSE: Your suggestion is much appreciated. We have divided mentioned section into two separately chapters: “Future perspectives” and “Conclusions”.
"Minor points for the text:
In lines 174, 180: Please remove the “s” from the word “ions” or the word “ions” completely since in the context it is not necessary."
RESPONSE: We have changed the fragment according to Reviewer’s suggestions.
"In line 183: Please rephrase to “In shoots, the mechanism...”.
RESPONSE: We have changed the fragment according to Reviewer’s suggestions.
"In line 188: Please remove the “s” from the word “genes”."
RESPONSE: We have done.
"In line 294: Please write “maintaining an optimal...”."
RESPONSE: We have changed the fragment according to Reviewer’s suggestions.
"In line 295: Please write “Na+” instead of “sodium”."
RESPONSE: We have done.
"Also in line 295: Please write “in the cytoplasm”."
RESPONSE: We have done.
"In line 323: Please write “underlying the unique innate immunity of plants”"
RESPONSE: We have done.
"In line 325: There are also intracellular PAMPs."
RESPONSE: Thank you for your valuable attention. We have included intracellular PAMPs mechanism in the appropriate fragment.
"In line 364: Please write “presumably a vital factor”."
RESPONSE: We have done.
"In line 372: Please write “the jasomnate (JA) signaling pathway”."
RESPONSE: We have done.
Round 2
Reviewer 2 Report
Dear authors,
you addressed all minor concerns. The new Figure is absolutley beautiful and greatly improves the article. Very well done. This article is ready for puplication and will be a great contribution to the field and to the scientific community.